# Investigating the origin and consequences of endogenous default options in repeated economic choices

**Joaquina Couto**[1]*, **Leendert van Maanen**[1,2,3ʘ], **Maël Lebreton**[2,4,5,6ʘ]

**1** Department of Psychology, University of Amsterdam, Amsterdam, the Netherlands, **2** Amsterdam Brain and Cognition (ABC), University of Amsterdam, Amsterdam, the Netherlands, **3** Department of Experimental Psychology, Utrecht University, Utrecht, the Netherlands, **4** Amsterdam School of Economics (ASE), University of Amsterdam, Amsterdam, the Netherlands, **5** Swiss Center for Affective Science, University of Geneva, Geneva, Switzerland, **6** Department of Basic Neurosciences, University of Geneva, Geneva, Switzerland

ʘ These authors contributed equally to this work.
* j.m.ferreiracouto@uva.nl

**Data Availability Statement:** All data files are available from the OSF database (DOI 10.17605/OSF.IO/TSJBU).

**Funding:** This work was supported by an EU Marie Sklodowska-Curie Individual Fellowship (IF-2015

## Abstract

Classical value-based decision theories state that economic choices are solely based on the value of available options. Experimental evidence suggests, however, that individuals' choices are biased towards default options, prompted by the framing of decisions. Although the effects of default options created by exogenous framing–such as how choice options are displayed–are well-documented, little is known about the potential effects and properties of endogenous framing, that is, originating from an individual's internal state. In this study, we investigated the existence and properties of endogenous default options in a task involving choices between risky lotteries. By manipulating and examining the effects of three experimental features–time pressure, time spent on task and relative choice proportion towards a specific option–, we reveal and dissociate two features of endogenous default options which bias individuals' choices: a natural tendency to prefer certain types of options (*natural default*), and the tendency to implicitly learn a default option from past choices (*learned default*). Additional analyses suggest that while the natural default may bias the standard choice process towards an option category, the learned default effects may be attributable to a second independent choice process. Overall, these investigations provide a first experimental evidence of how individuals build and apply diverse endogenous default options in economic decision-making and how this biases their choices.

## Introduction

Consider the choice between an investment involving a low amount of money but a high chance of getting that amount versus an investment involving a high amount but a low chance of getting that amount. Classical decision theories state that, when making such a choice, individuals aggregate attributes of available options (i.e., probabilities and amounts) into a decision

Grant 657904) and a NWO Veni (Grant 451-15-015) awarded to ML. ML also acknowledge the support of the Bettencourt-Schueller Foundation through the Prix Jeunes Chercheur 2014. JC is supported by a PhD fellowship from the FCT (SFRH/BD/132089/2017).

**Competing interests:** The authors have declared that no competing interests exist.

variable ("value") and select the highest valued option [1,2]. Under this value-based decision-making framework, choices were initially assumed to be based solely on the value of the options [3,4]. This assumption has, however, been challenged by robust evidence suggesting that choices often depend on apparently irrelevant factors. For instance, individuals' choices have been shown to be biased towards default options, induced by the way the options and decision problems are framed [5,6]. Extensive research has evidenced the strength and pervasiveness of such biases, created by exogenous–external–framing effects in our daily life: dramatic behavioral differences in the choice to save money towards retirement [7] or to be an organ donor [8] have been reported, depending on the decision being presented as "opting-out" or "opting-in". A recent meta-analysis of 58 studies further reveals that the effects of exogenous default-options are highly replicable, with moderate-to-high effect sizes [9]. On the other hand, less is known about how individuals internally frame decisions and build default options in the absence of such exogenous cues (e.g., the tendency to choose the same seats, even without a seating chart), and how these potential *endogenous* default options bias choices [10]. Endogenous default options are, thus, based on the same core idea as the exogenous default options (i.e., the tendency to stick with a default-option and, thereby, avoid alternative-options), with the substantial difference that in the exogenous case, the default options are explicitly framed to individuals (e.g., as "opting-in" or "opting-out"), not internally framed and built by individuals. In this respect, endogenous default options are very similar to a priori biases studied, for example, in the perceptual decision literature [11,12].

Yet, endogenous default options are a cornerstone of decision theories in behavioral economics [13,14]. Similarly, endogenous default options are at the heart of a broad class of *dual-process theories* of cognition, which have been extremely influential in cognitive and social sciences [15,16]. Under this dual-process framework, judgments and choices are assumed to be *endogenously*–internally–framed as a default option versus alternative options, with the default option being preferentially endorsed and chosen by a fast and automatic process, and the alternative options by a slow, deliberative one [17]. In order to investigate dual-process theories experimentally, a number of studies have attempted to leverage the presumed difference in processing speed between the two processes, assuming that options that are chosen under time pressure represent a *natural default option*, endorsed by a fast and automatic process [18]. This reasoning has been applied in various types of decision in behavioral economics or behavioral ethics, to investigate whether individuals are naturally inclined to be, for example, risk-averse vs risk-seeking [19,20] or altruistic vs egoistic [21,22]. However, the simple inference that fast choices relate to a natural default option is quite problematic: indeed, without very strong control on key experimental variables and features, simple analyses of response times (RTs) can be susceptible to critical confounds and misinterpretations such as inappropriate reverse inferences [23]. On the other hand, time pressure manipulations and RT analyses are still valuable tools to investigate endogenous default option biases, when properly implemented (i.e., with careful control over multiple experimental factors, or thorough model-based RT analyses) [24]. Such rigorous experimental designs and/or RT analyses may help to understand the inconsistent experimental results that have been reported in the field, and to ascertain whether time pressure increases risk-aversion vs. risk-seeking, or altruism vs. egoism, which is still highly debated [25–27].

In the present study, we aimed at providing evidence for the existence of endogenous default option biases in economic choice tasks. We define three non-mutually exclusive types of endogenous default options, which are induced and/or modulated by different experimental factors. First, in line with previous studies in behavioral economics [19–22], we define *natural default options*, which are hard-wired to be endorsed by a fast and automatic process and which are increasingly chosen under increasing time pressure. Second, we define *dominant*

*default options*, induced by the mere repetition of the most frequent choice type, and which are increasingly chosen under increasing time pressure (habit) [28–30]. Finally, we define *learned default options*, which are increasingly induced over time by the repetition of the most frequent choice type, regardless of time pressure (passive learning) [31–33]. It should be noted that repetition of the most frequent choice type in both dominant and learned default options refers to the repetition of individuals' own choices and not to stimuli repetition. Importantly, all these types of endogenous default options, although potentially underpinned by different neuro-cognitive processes, could be confounded under simple experimental designs, and lead to mis-interpretations and apparently contradictory results.

Experimentally, we designed several adaptations of a standard economic task involving binary choices between lotteries with varying monetary amounts and probabilities of gain (gain version) or loss (loss version). We carefully controlled and/or orthogonalized several key experimental factors (time pressure, time on task, most frequently chosen option) so as to ulti-mately test the presence of the three proposed endogenous default options in risky choices, based on simple pattern of choices (i.e., risk-averse vs. risk-seeking) observed when those fac-tors were manipulated. We further investigated the mechanisms at stake in the emergence of these endogenous default option biases and proposed two competing mechanistic hypotheses: a mixture hypothesis which–in line with dual-process theories–, assumes that biases are caused by changes in the balance between an automatic and a deliberative process vs. a bias hypothesis proposing that such biases are rather caused by computational changes within a single process [34]. We assessed the relative merits of these hypotheses by quantifying how RT distributions associated with the experimental manipulations mimic properties of binary mixtures under different mixture proportions, capitalizing on the so-called fixed-point property of binary mix-ture distributions [35,36]. The fixed-point property refers to the fact that different mixture proportions of the same two base distributions share a common probability density point. In the present study, this feature can be used as a statistical test of a quite conservative specifica-tion of dual-process hypothesis: if we observe a fixed-point for RT distributions observed under different experimental conditions, then RTs are likely to be generated by two indepen-dent processes, whose mixture proportion was modulated by the experimental manipulation.

## Three non-mutually exclusive types of endogenous default options

We hypothesize that the various types of endogenous default options are not mutually exclu-sive. Yet, specific experimental manipulations may elicit effects that can only be attributed to a subset, hence could be used to disambiguate the effects caused by different types of endoge-nous default options. **Fig 1** schematically illustrates how the hypothesized default options affect choice behavior under the following experimental manipulations: time pressure, time spent on a task, and relative choice proportion towards one specific option. In contrast to time pressure, which has been typically used in studies from behavioral economics to reveal endogenous default options [19,20], the two last factors have often been neglected and/or not analyzed in the field. Importantly, they could explain the inconsistencies between results of different stud-ies. In the absence of any hypothesized default options, time pressure and time on task are pre-dicted to have no effect (**Fig 1A**). In the presence of *natural default option* [19–22], time pressure unidirectionally changes the proportion of choices–either resulting in a higher or lower proportion of safe choices (**Fig 1B**, top panel). This is independent of the relative choice proportion: i.e., the effects are identical regardless of the fact that participants might choose more often a specific lottery type. Also, the effects of natural default option are independent of any learning or habituation process. Therefore, in the sole presence of natural default option, there should be no effect of the time spent on a task (**Fig 1B**, bottom panel). In the presence of

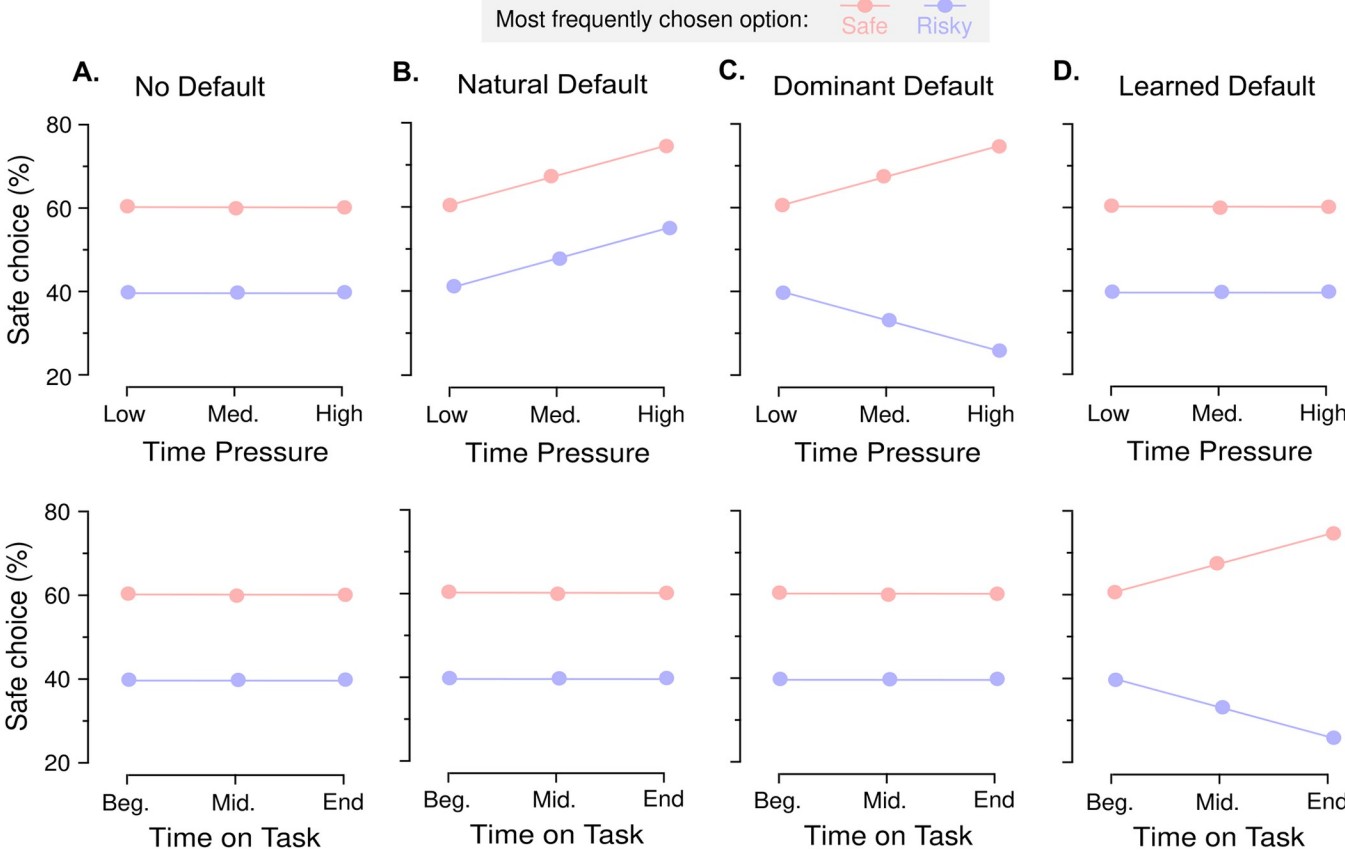

**Fig 1. Hypotheses about the underlying causes of default options.** Panels depict, under different hypotheses the expected effect of time pressure (top) and time on task (bottom) on the choice pattern of two typical individuals, who predominantly choose the safe (red dots) or the risky (blue dots) option. **A.** No default option. Neither time pressure nor time on task facilitates the choice of a default option. **B.** Natural default option. Time pressure may facilitate the choice of natural default options. Under this hypothesis, effects of time pressure do not depend on the manipulation of choice proportion, so individuals' choices are naturally biased towards an option category (here, the safe lottery), regardless of the most frequently chosen option. **C.** Dominant default option. Time pressure may, instead, facilitate the choice of dominant default options. Under this hypothesis, effects of time pressure depend on the manipulation of choice proportion, so individuals' choices are biased towards the most frequently chosen option, by means of simple repetition. **D.** Learned default option. Time on task may facilitate the choice of learned default options. Effects of time on task also depend on the manipulation of choice proportion, so individuals' choices are biased towards the most frequently chosen option.

*dominant default option*, time pressure changes choice proportion so as to amplify the selection of the type of options that is already predominantly chosen (**Fig 1C**, top panel) [28–30]. Therefore, and as opposed to the natural default option case, the effects of time pressure should depend on the relative choice proportion. Yet, similar to the natural default option hypothesis, the effects of dominant default options are independent of any learning or habituation process (**Fig 1C**, bottom panel). In the presence of *learned default option*, previously chosen options are increasingly chosen, as a function of the number of past choices–i.e. time spent on the task (**Fig 1D**, bottom panel). These effects are presumed to be driven by passive learning from participants' past choices (i.e., which option they choose most throughout the task, regardless of time pressure) [31–33]. In the presence of *learned default option*, the effects of time on task should therefore depend on the average choice proportion. Note that other factors can also create other specific patterns of choice (see **S1 File**, **Additional Behavioral Predictions**). Yet, because those patterns are not consistent with the results of a preliminary experiment (see **S1 File**, **Supplementary Experiments**), we did not include those in our main default option hypotheses.

We tested these hypotheses in two different contexts (i.e., gain and loss tasks), for which we orthogonally manipulated the three experimental factors (time pressure, time spent on task, and average choice proportion). Average choice proportion refers to which option (risky or safe) is predominantly chosen by participants, in a specific instantiation of the task and in the absence of other effects. We manipulated this factor in a between-subject design to specifically assess whether time pressure truly increases natural default option choices or just facilitates dominant choice repetition (natural vs. dominant default hypotheses).

## Experiment 1

### Methods

**Participants.** The experiment was approved by the local Ethics Committee of the University of Amsterdam and all participants gave a written informed consent prior to taking part in the study.

37 participants enrolled and completed the experiment (27 females, mean age = 26.5, SD = 11.9; targeted sample size = 36). A sample size of 36 ensures enough power to detect an effect, given effect sizes in comparable experiments [9] and the fact that using generalized linear mixed models (see **Statistical Analysis**) [37] requires substantially smaller sample sizes than e.g., analysis of variance. Participants were paid according to a performance-based payment incentive, rather than a purely flat-fee payment [38]. They had the opportunity to choose between a course credit (1 ECTS) or a financial compensation (base amount of 7€), with the possibility of getting an extra amount of 4€, depending on a randomly chosen trial. The conversion rates between experimental euros (EE)/real euros (€) were set such that the highest outcome lottery faced by the participants scaled to a 4 real € gain. This led to a conversion rate of approximatively 15 EE/€. Participants could gain an extra 1€ if they complied with the time pressure instructions.

**Design.** The experiment was programmed in Cogent for Matlab®. We designed a repeated binary decision-making task involving probabilistic monetary outcomes, framed as potential gains (**Fig 2A**). At each trial, participants were presented with a wheel-of-fortune which defined two options. The option featured as the Safe lottery offered a probability p > 50% (e.g., 75%) of winning a certain amount of money a (e.g., 4€), and the option featured as the Risky lottery offered a probability 1-p (e.g., 25%) to win a higher amount A (e.g., 8€). The probabilities were presented as the two complementary areas of the wheel-of-fortune and the amounts as vertical bars of varying height. The bars were up relative to the center of the screen. The attributes of the risky and safe lotteries were colored in blue or yellow and this color attribution was balanced across participants. The side of presentation (left or right) of the safe and risky lotteries was also balanced across trials. Text describing exact probabilities and amounts was presented on the same screen.

The experiment involved three phases–a training session, a priming session and an experimental session (**Fig 2B**). During the training session, participants experienced 10 trials with feedback, to familiarize with the task and the probabilities: once participants selected their preferred option, the wheel-of-fortune was spun, and participants could win the amount paired with the chosen option if the spin stopped on the chosen-option portion. After the pointer was rotated, explicit feedback was given (e.g., "You win a €!"). In order to induce the effects of the average choice proportion manipulation from the first trial of the task, participants next performed 72 trials with no feedback and in a self-paced mode, during a priming session. Participants were assigned to two conditions: safe default and risky default conditions. Participants assigned to the safe default condition were presented with more desirable *Safe* lotteries, such that the expected probability of choosing the *Safe* lottery was higher than the *Risky* lottery. Conversely, participants assigned to the risky default condition were presented with more

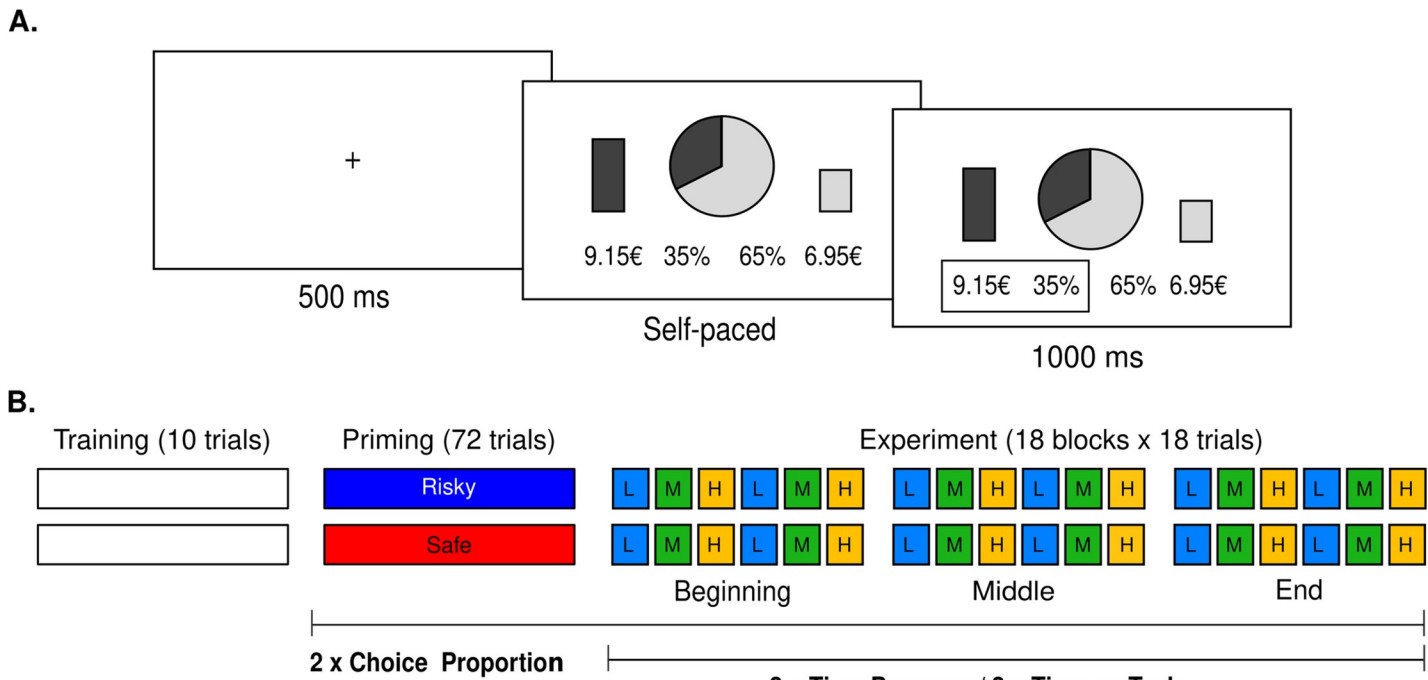

**Fig 2. Experiment. A.** Behavioral task. Successive screenshots displayed during a given trial are illustrated from left to right, with durations in milliseconds. At each trial, following a short fixation (500ms), subjects have to choose between a risky (here, left: 35% chance to win 9.15€) and a safe (here, right: 65% chance to win 6.95€) lottery. Choices are self-paced, and followed by a choice-confirmation screen, where the selected lottery is highlighted by a contour box. **B.** Experimental Design. Subjects start with a training session to familiarize themselves with the task. In the priming (72 trials) and experimental (18 even blocks of 18 trials) sessions, three experimental factors are orthogonally manipulated: time pressure, time spent on task, and average choice proportion. Average choice proportion is manipulated between-subjects in both the priming and experimental sessions, by designing different sets of lotteries (see Methods). Time pressure and time on task are manipulated within-subjects in the experimental session, in a counterbalanced way, such that all three levels of time pressure are experienced (2 blocks of 18 trials) at every stage of the task (beginning, middle end).

desirable *Risky* lotteries, such that the expected probability of choosing the *Risky* lottery was higher than the *Safe* lottery (70% on average, respectively). The experimental session comprised 18 blocks of 18 trials. These 18 blocks were composed of 6 repetitions of 3 time pressure conditions: low, medium, and high time pressure. Participants were presented with cues on which condition to follow in the beginning of each trial: THOUGHTFUL required participants to be "as thoughtful as possible", THOUGHTFUL and FAST required them to be "as thoughtful and as fast as possible", and FAST to be "as fast as possible". The order of time pressure conditions was balanced within participants, such that all three levels of time pressure were orthogonal to the time spent on task. In other words, all three levels of time pressure were experienced (2 blocks of 18 trials) at every stage of the task (beginning, middle, and end) in a different order. At the end of the experiment, one of the participants' choices was selected and executed, and the resulting gains were added to their payoffs. To incentivize compliance with the time pressure instructions, two choice RTs–corresponding to two different time pressure conditions–were selected and compared too. Participants gained an additional bonus of 1€ if the RTs of these two trials differed in the instructed direction (e.g., shorter RT in the high time pressure condition than in the low time pressure condition). All the payment procedures were explained in detail before the experiment.

**Stimuli.** An exhaustive description of the stimulus generation procedure is available in the **S1 File, Stimuli Generation.** We designed an original model-fitting followed by a model-inversion approach to generate stimuli (i.e., lottery pairs), so as to operate a tight control over the options

that our participants may choose–*in expectations*. Briefly, and as is common practice in the field, we assumed that participants' probability $p_s$ to choose the safe option derives from a softmax on the difference in expected utility between the safe and the risky option ($\Delta EU$) [39,40]:

$$p_s = 1/(1 + \exp(-\boldsymbol{\beta} \times \Delta EU)); \text{ where} \tag{1}$$

$$\Delta EU = EU_{safe} - EU_{risky} = P \times a^{\mathbf{u}} - (1 - P) \times A^{\mathbf{u}}. \tag{2}$$

This model has two free-parameters, $\boldsymbol{\beta}$ and $\mathbf{u}$, which respectively represent the inverse temperature (i.e. how stochastic vs. utility maximizing choices are), and the utility curvature (i.e., how risk-seeking vs. risk-averse choices are). $P$ and $a$ refer to the probability and amount associated with the Safe lottery, while *1-P* and *A* to the probability and amount associated with the Risky lottery. It should be noted that this model does not take into consideration probability distortions [41], as probabilities in our stimuli set (both *P* and *1-P*) are middle range (20–80)– i.e. in the 'linear' region of the inverse S-shape form. Probability distortions would typically be observed on the extremes of the probability range, which we did not measure.

The values for $\boldsymbol{\beta}$ and $\mathbf{u}$ that we used to generate stimuli were based on the averaged individual estimates from the preliminary experiment (see **S1 File**, **Supplementary Experiments**). Model parameters were estimated at the individual level by minimizing free energy [42,43], using a variational Bayes approach under the Laplace approximation [44]. In order to avoid any bias in the parameter estimation, we only used data from a no time pressure block (216 choices). We used wide, unbiased priors (mean (±SD): $\boldsymbol{\beta}$ = 3(10); $\mathbf{u}$ = 1(10)).

The results (mean (± STD)) of the estimation procedure were $\boldsymbol{\beta}$ = 4.38 (3.65) and $\mathbf{u}$ = 0.56 (0.54) and the mean values were used to generate stimuli for the safe default and risky default conditions. Specifying *P* and *a*, we can solve for *A*:

$$A = \left( \frac{\frac{1}{\boldsymbol{\beta}} \times \log(\frac{p_s}{1-p_s}) + P \times a^{\mathbf{u}}}{1 - P} \right)^{1/\mathbf{u}} \tag{3}$$

The manipulation of choice proportion then simply consisted in designing sets of choices (*A*, *a*, *P*) while setting $p_s$>50% on average for the safe default condition group (we chose $p_s$ = 70%), and $p_s$<50% on average for the risky default condition group (we chose $p_s$ = 30%).

**Statistical analyses.** To comprehensively test for the effects of time pressure, time spent on task, and choice proportion, we designed and compared several generalized linear mixed models (GLMMs). GLMMs were performed with the lmerTest package [45], an extension of lme4 package [46] in R. We used a linear link function when modelling RTs (just for manipulation check purposes), and a logit link function when modelling choices. Choices were coded as 1 for safe and 0 for risky. The independent variables were time pressure (TP, three levels: Low, Medium, High), time spent on task (ToT, three levels: Beginning, Middle, End), and choice proportion (CP, two levels: Safe and Risky). Independent variables with more than two levels (TP and ToT) were coded as cardinal–i.e., as continuous, linear variables (taking values -1, 0 and 1 for increasing levels of TP and ToT). All models included individual random effects (random intercepts and slopes) for the independent variables TP and ToT.

We used a backward iterative model-comparison approach [47]: we started from the most complex model, which included all possible interactions between the explanatory variables, and iteratively removed non-significant interactions and subsequent non-significant variables. At each iteration, we performed a likelihood-ratio test between the new (simpler) and former (more complex) models, to assess whether the additional interactions and/or variables included in the former model were statistically warranted by the data. This procedure was

stopped when the test came out significant, i.e., when removing an interaction and/or variable did significantly decrease the goodness-of-fit (see **S1 File**, **GLMMs**).

## Results

To test for the success of our manipulation of choice proportion, we first analyzed the behavior observed in the priming session. Results show that participants predominantly chose the option designed to be the most frequently chosen (**Fig 3**). That is, participants assigned to the safe default condition predominantly chose safe options (safe choice proportion: 67% ± .06; one-sample t-test against 50%; $t_{18}$ = 2.76, p = .006485); and participants assigned to the risky default condition, predominantly chose risky options (safe choice proportion: 31% ± .07; one-sample t-test against 50%; $t_{17}$ = -2.85, p = .005489). Note that the observed probabilities of choosing in both groups are very close to the expected probabilities (i.e., 70% of safe choices in the safe default condition group and remaining 30% in the risky default condition group).

**Dissociable effects of time pressure and time spent on task on default bias.** We first analyzed RTs from the experimental session and the optimal model suggests that our manipulation of time pressure was successful: participants successfully decreased their RTs under increasing time pressure (see **S1 File**, **GLMMs**).

As for choices, the optimal GLMM included main effects for our three experimental factors ($\beta_{TP}$ = .22, SE = .09, p = .010781; $\beta_{ToT}$ = .15, SE = .08, p = .058333; $\beta_{CP}$ = -2.20, SE = .63, p < .001 –see **Fig 3A**), and one significant interaction term between the time on task and the choice proportion manipulation ($\beta_{ToT \times CP}$ = -.31, SE = .12, p = .009173 –see **Fig 3B**). These results indicate that participants modulated their choice pattern according to our three types of manipulation. Regarding the manipulation of choice proportion, participants predominantly chose the option designed to be the most frequently chosen. Regarding the manipulation of time pressure, participants increasingly chose the safe option with increasing time pressure, regardless of the most frequently chosen option. Finally, regarding the manipulation of time on task, participants increasingly chose the most frequently chosen option while they progressed through the experiment.

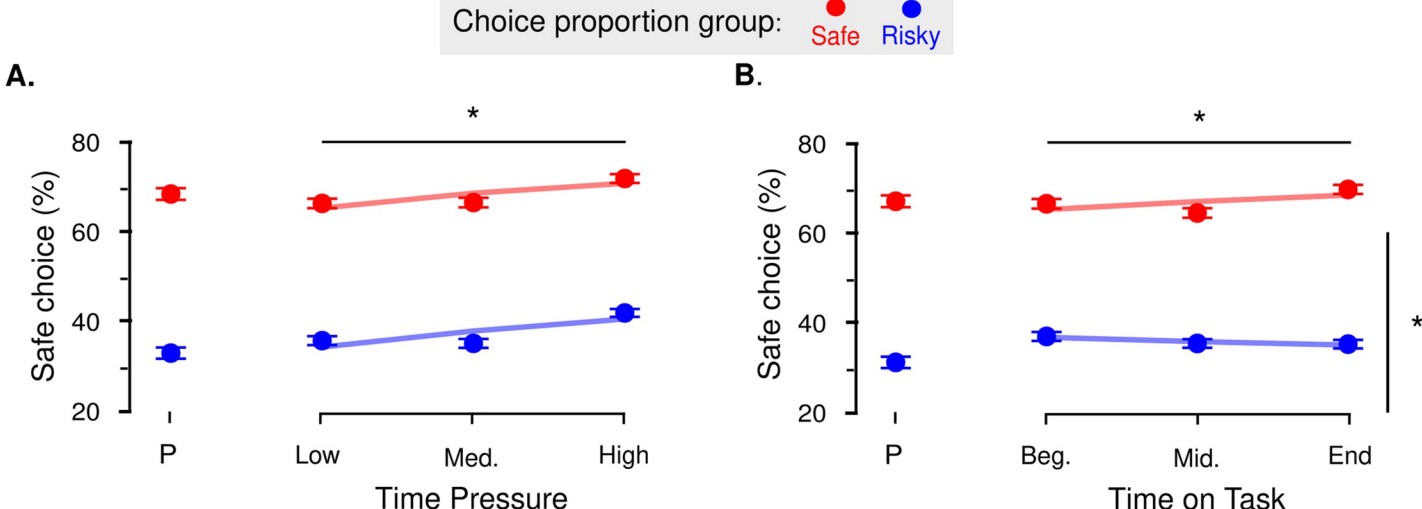

**Fig 3. Observed proportion of safe choices. A.** Time Pressure. Regardless of the most frequently chosen option, participants increasingly chose the safe lotteries. **B.** Time on Task. In the two groups, participants also learned with time to choose increasingly more the most frequently chosen option: Red and blue dots indicate data from the two groups of participants, where the safe (respectively risky) option is designed to be most frequently chosen. Dots and error bars indicate sample mean and within-subject standard error of the mean. Lines represent the average fit of the generalized linear mixed-effects regressions. *: P<0.05 for the main effect of time pressure and the interaction between choice proportion and time on task. P: priming session.

## Discussion

These results uncovered two features of endogenous default options which bias individuals' choices in gain contexts. First, increasing time pressure revealed the existence of a natural default option, regardless of the choice proportion or time on task. Our results suggest that, under time pressure, individuals are naturally biased towards the safe option when facing potential gains. Second, an interaction between the choice proportion manipulation and the time on task revealed the existence of a learned default option. Here, our results show that, while progressing through the experiment, individuals exhibited an increased tendency to choose the option that was most frequently chosen. Importantly, they evidence the importance in dissociating non-mutually exclusive (and potentially confounded) factors with a rigorous experimental design (and through a careful control over such factors). To check if these endogenous default options effects generalize to other exogenous default option contexts, we designed a second experiment involving a loss lottery framing.

## Experiment 2

### Methods

**Participants.** The experiment was approved by the local Ethics Committee and all participants gave a written informed consent prior to taking part in the study.

36 participants enrolled and completed the experiment (28 females, mean age = 23.2, SD = 6.7; targeted sample size = 36). Again, a sample size of 36 ensures enough power to detect an effect [9,37]. As in Experiment 1, participants were paid according to a performance-based payment incentive [38]. They had the opportunity to get a financial compensation (base amount of 12€), with a potential subtraction from this compensation of 4€ depending on a randomly chosen trial. The conversion rates between experimental euros (EE)/real euros (€) were set such that the highest outcome lottery scaled to a 4 real € loss, leading to a conversion rate of approximatively 50 EE/€. Participants could also gain an extra 1€ if they complied with the time pressure instructions.

**Design.** Experiment 2 was similar to Experiment 1, and consisted of a repeated binary decision-making task involving probabilistic monetary outcomes, framed as potential losses (**Fig 4**). The option featured as the Safe lottery offered a probability $p > 50\%$ (e.g., 75%) of losing a certain amount of money $a$ (e.g., 4€), and the option featured as the Risky lottery offered

**A.**

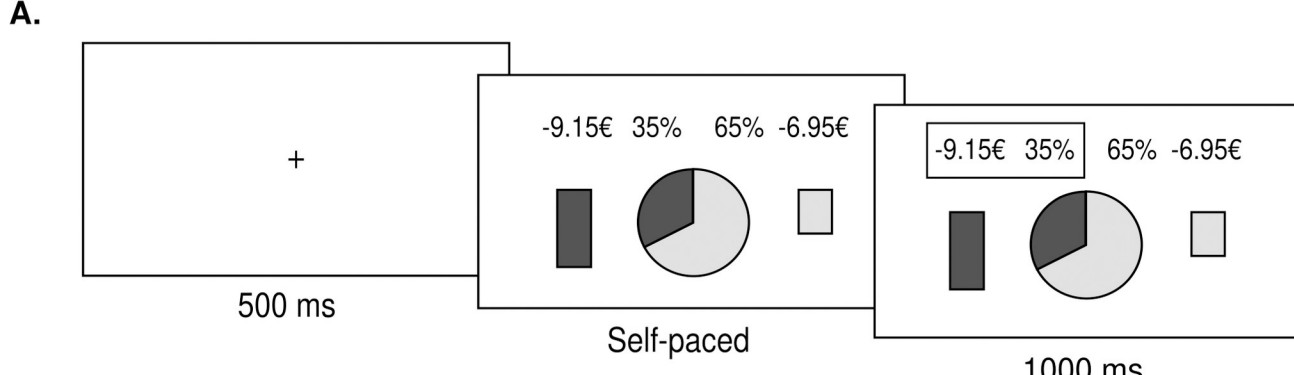

**Fig 4. Experiment. A.** Behavioral task. Successive screenshots displayed during a given trial are illustrated from left to right, with durations in milliseconds. At each trial, following a short fixation (500ms), subjects have to choose between a risky (here, left: 35% chance to lose 9.15€) and a safe (here, right: 65% chance to lose 6.95€) lottery. Choices are self-paced, and followed by a choice-confirmation screen, where the selected lottery is highlighted by a contour box.

a probability *1-p* (e.g., 25%) to lose a higher amount *A* (e.g., 8€). The probabilities were presented as the two complementary areas of the wheel-of-fortune and the amounts as vertical bars, down relative to the center of the screen. Color attribution of the safe and risky lotteries was balanced across participants and the side of presentation was also balanced across trials. Text describing exact probabilities and amounts was presented, and the amounts were displayed with a minus sign.

The experiment also involved three phases–a training session, a priming session and an experimental session (**Fig 2B**). During the training session, participants experienced 10 fictive trials with feedback, to familiarize with the task and the probabilities. Participants next performed 72 trials with no feedback and in a self-paced mode, during a priming session. Similar to Experiment 1, participants were assigned to two conditions: safe default and risky default conditions, for which a certain probability of choosing a specific option was prompt (i.e., the Safe lottery in the safe default condition group, and the Risky lottery in the risky default condition group). The experimental session comprised 18 blocks of 18 trials, and the 18 blocks were composed of 6 repetitions of 3 time pressure conditions: low, medium, and high time pressure. The order of time pressure conditions was balanced within participants, such that all three levels of time pressure were orthogonal to the time spent on task. At the end of the experiment, one of the participants' choices was selected and executed, and the resulting losses were subtracted to their payoffs. Two choice RTs–corresponding to two different time pressure conditions–were selected and compared too. Participants gained an additional bonus of 1€ if the RTs of these two trials differed in the instructed direction. All the payment procedures were explained in detail before the experiment.

**Stimuli.**   Stimulus generation procedure is the same as for Experiment 1. The whole procedure relies on an estimation–and subsequent inversion–of the expected utility model used by participants in a preliminary experiment (see **S1 File**, **Supplementary Experiments** [39,40]). Based on the expected utility model, the idea is to generate stimuli (i.e., lottery pairs), whose attributes prompt a certain probability of choosing a specific option (i.e., the Safe lottery in the safe default condition group, and the Risky lottery in the risky default condition group). The procedure only differs from the procedure in Experiment 1 in the values of the amounts (*a* and *A*): while the values of *a* and *A* are positive in the in Experiment 1 (gain version), they are negative in Experiment 2 (loss version)—see **S1 File**, **Stimuli Generation** for a complete description of the procedure.

**Statistical analysis.**   To comprehensively test for the effects of time pressure, time spent on task, and choice proportion, we designed and compared several GLMMs. Similarly to Experiment 1, we used a linear link function when modelling RTs (again, just for manipulation check purposes), and a logit link function when modelling choices. Choices were coded as 1 for safe and 0 for risky. The independent variables were time pressure (TP), time spent on task (ToT), and choice proportion (CP). TP and ToT were coded as continuous, linear variables (taking values -1, 0 and 1). All models included individual random effects (random intercepts and slopes) for TP and ToT.

As in Experiment 1, we used a backward iterative model-comparison approach [47]: we started from the most complex model, which included all possible interactions between the explanatory variables, and iteratively removed non-significant interactions and subsequent non-significant variables, until the test came out significant, i.e., when removing an interaction and/or variable did significantly decrease the goodness-of-fit (see **S1 File**, **GLMMs**).

## Results

We first analyzed the behavior observed in the priming session. Results show that participants predominantly chose the option designed to be the most frequently chosen (**Fig 5**). In this

case, participants assigned to the safe default condition predominantly chose safe options (safe choice proportion: 54% ± .07, one-sample t-test against 50%; $t_{17}$ = .51, p = .3074); and participants assigned to the risky default condition, predominantly chose risky options (safe choice proportion: 30% ± .07, one-sample t-test against 50%; $t_{17}$ = -2.98, p = .004159). Although the observed probabilities are not so close to the expected probabilities in the loss task (especially in the safe default condition), the observed probabilities are still very satisfactory.

RTs were first analyzed. Although the optimal model differed from the one in Experiment 2, its conclusion regarding the manipulation control go in the same direction: participants successfully decreased their RTs under increasing time pressure (see **S1 File, GLMMs**).

As for choices, like for Experiment 1, the optimal GMLER model included significant main effects for our three experimental factors ($\beta_{TP}$ = -.28, SE = .09, p = .00169; $\beta_{ToT}$ = -.36, SE = .14, p = .009023, $\beta_{CP}$ = 1.73, SE = .58, p = .002921 –see **Fig 5A**), and one significant interaction term between the time on task and the choice proportion manipulation ($\beta_{ToT \times CP}$ = .46, SE = .18, p = .010607 –see **Fig 5B**). Again, these results indicate that participants modulated their choice pattern according to our three types of manipulation: participants predominantly chose the option designed to be the most frequently chosen (choice proportion manipulation); participants increasingly chose the risky option with increasing time pressure, regardless of the most frequently chosen option (time pressure manipulation); and participants increasingly chose the most frequently chosen option while they progressed through the experiment (time on task manipulation).

## Discussion

Altogether, these results form a coherent pattern, which replicates in the two different frames of Experiment 1 and 2 (gain and loss tasks, respectively). Specifically, under time pressure, individuals are naturally biased towards the risky option when facing potential losses, regardless of the choice proportion or time on task manipulations (natural default option). While progressing through the experiment, individuals also exhibited an increased tendency to choose the option that was most frequently chosen (learned default option).

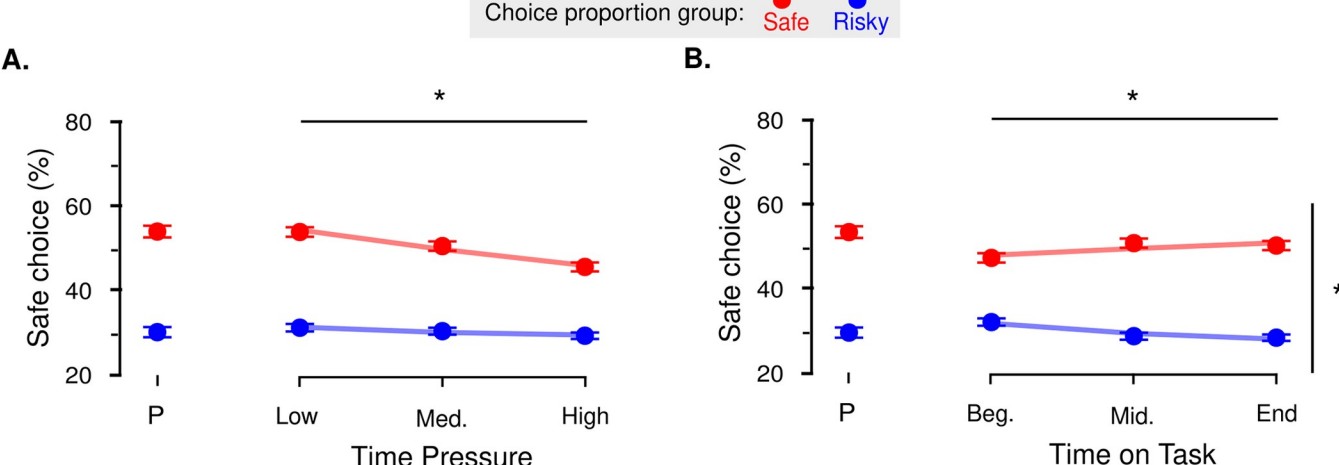

**Fig 5. Observed proportion of safe choices. A.** Time Pressure. Regardless of the most frequently chosen option, participants increasingly chose the risky lotteries. **B.** Time on Task. In the two groups, participants also learned with time to choose increasingly more the most frequently chosen option: Red and blue dots indicate data from the two groups of participants, where the safe (respectively risky) option is designed to be most frequently chosen. Dots and error bars indicate sample mean and within-subject standard error of the mean. Lines represent the average fit of the generalized linear mixed-effects regressions. *: P<0.05 for the main effect of time pressure and the interaction between choice proportion and time on task. P: priming session.

## Hypotheses on endogenous default options mechanisms

Although the results of Experiment 1 and 2 clearly constitute evidence for the concurrent existence of two main endogenous default option dimensions which bias individuals' choices (natural and learned), questions remain about the mechanisms at stake in the emergence of these default option biases. More specifically, two types of mechanistic hypotheses can be proposed. A *mixture hypothesis* by dual-process theories proposes that the balance between two independent processes in charge of decision-making changes as participants are under time pressure or due to the time participants spend on a task [15,16]. Alternatively, a *bias hypothesis* proposes that a single process is in charge of the decision, but the specific computations are influenced by the time pressure or the time on task [34]. In order to test these competing hypotheses, we analyzed whether the RT distributions of Experiment 1 and 2 showed a common density point, the so-called fixed-point property [35,36,48]. The fixed-point property exists when the observed data are the result of a mixture of two base distributions, with various mixture proportions for separate conditions. In that case, the observed RT distributions all cross at a common density point, regardless of the mixture proportion (**Fig 6A**). Observing a fixed-point in the RT distributions across the different conditions of time pressure and/or time on task would then be evidence for the *mixture hypothesis* as a mechanistic cause of the natural default and/or learned default bias.

## Methods

The fixed-point property was estimated and tested with the fp package in R [35], available at http://www.leendertvanmaanen.com/fp), which proceeds in three steps. In step 1, RT distributions for each participant and each condition (i.e., different levels of time pressure and time on task) are estimated using a Gaussian kernel-based density estimator. This means that based on the observed data pattern, a smooth estimate is selected that summarizes how the behavior is distributed under the three conditions of time pressure and time on task. Following the recommendations by [35] the kernel standard density was set equal to the standard deviations of the RT data, collapsed across all conditions. In step 2, for each pair of density functions, the RTs of the crossing points are computed. Here, given that we have 3 levels per experimental manipulation (TP: Low,

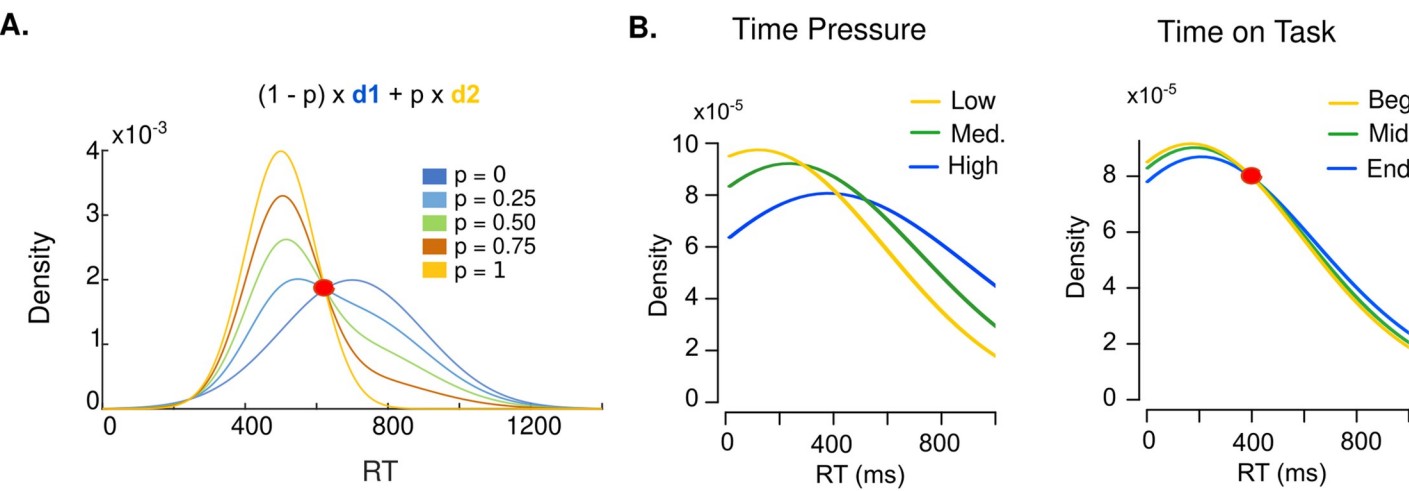

**Fig 6. Fixed-point property of binary mixtures holds for time on task but not for time pressure. A.** Theory. The fixed-point property entails that any mixture of two base distributions (here, $d_1$ and $d_2$) cross at the same common density point, regardless of the mixture proportion ($p$). Densities with various mixture proportions of $d_1$ and $d_2$ are displayed The red dot indicates the fixed-point. **B.** Smoothed histograms of response time distributions elicited by the different levels of time pressure (left) and time on task (right). Fixed-point analysis (see main text and Methods) suggest that the response time distribution elicited by the different levels of time pressure do not cross the same common density point (left), whereas the response time distributions elicited by the different levels of time on task do (right).

Medium, High; ToT: Beginning, Middle, End), this results in 3 pairs of density functions (e.g., TP: Low—Medium, Low—High, Medium—High), hence 3 crossing points per participant. In step 3, analysis of variance is used to determine whether, across participants these three samples of crossing points are likely to come from multiple populations–which is evidence against a fixed-point -, or whether they are more likely to be drawn from the same population–which is in favor of a fixed-point. Because traditional null hypothesis significance testing typically does not quantify support for a null hypothesis (but only indicates whether the data is unlikely to occur under the null hypothesis), we used a Bayesian statistical analysis instead [49].

Because the hypothesis that we aim to test (mixture vs bias) should explain our results in both gains and loss contexts, the fixed-point property test should yield similar results regardless of the direction of the investigated frame (i.e., gain and loss frame). We therefore aggregated data from both experiments (Experiment 1 and 2).

## Results

Following the fixed-point analysis rationale, we quantified the probability that the crossing points of the RT distributions computed across the different conditions of time pressure and time on task were drawn from a single underlying population–i.e., are noisy samples of a fixed-point–or drawn from multiple populations, using Bayesian statistics [49]. We found no evidence in favor of the fixed-point property in the RT distributions elicited by the different conditions of time pressure (BF = .12; indicating that the data is 8.3 times more likely to be generated by multiple crossing points under time pressure manipulations than by one single crossing point, **Fig 6B, left**). However, the same analysis applied to the RT distributions elicited by the different conditions of time spent on task point towards the likely presence of such a fixed-point (BF = 3, **Fig 6B, right**).

As a negative control, we also tested the presence of the fixed-point property across time pressure conditions in the preliminary experiment (see **S1 File**, **Supplementary Experiments**). Since the effects of time pressure and time spent on task were confounded in that experiment, a fixed-point should not be observed [36]. Confirming our rationale, we did not find evidence in favor of the fixed-point property in this dataset (BF = .65 indicating the data is 1.5 times more likely to be generated by multiple crossing points than by one single crossing point).

## Discussion

The results of the fixed-point property analysis indicate that the natural default option bias results from a distortion of the decision-making process under time pressure (in line with the bias hypothesis), but that the learned default option bias results from the emergence of an independent process endorsing the most frequently chosen option with increasing time on task (in line with the mixture hypothesis). Although a Bayes factor of 3, as we found, is not typically considered strong [50], the fixed-point test is a conservative test [36]. Hence, the results in favor of the fixed-point property for different levels of time on task convincingly support the mixture hypothesis. Interestingly, though, we do not exclude the possibility that the results of not observing the fixed-point property for the different levels of time pressure could also be interpreted in light of such conservative assumptions. That is, if time pressure affects some other aspect of the data in addition to a change in the mixture proportion, the common crossing point may not be found (as we expected for the negative control experiment [36]).

## General discussion

With this study, we provide new experimental evidence of how individuals build and apply endogenous default options in economic decision-making and how this biases their choices.

The results consistently demonstrate that, under time pressure, individuals' choices are increasingly biased towards a default option category–the safe or risky option–, which depends on the framing of lotteries–potential gains or losses, respectively. The observation that time pressure has opposite effects on risky-choice proportion in the two frames rules out the possibility that these effects are due to non-value-based strategies or heuristics (e.g., increasingly choosing the option with the larger filled area on the lottery pie chart with increasing time pressure), and rather indicates that the observed behavior is a reflection of economic judgments. Although an increasing number of research has investigated the effects of time pressure on individuals' risk behavior, there is still no consensus about its effects in the two frames. In line with our findings, some studies report an increase of risk-aversion in the gain domain and risk-seeking in the loss domain [51]. Yet, other studies report the opposite choice pattern [52], or even no effect for one of the domains [53,54]–for more details, see the meta review table in [25]. Here, we report that time pressure increases risk-aversion in the gain domain and risk-seeking in the loss domain, a result which is consistent with two recent studies leveraging large sample sizes [19,55]. These converging pieces of evidence therefore suggest that time pressure increases the valence-dependent pattern of risk preference posited by Prospect Theory [13]: in contexts were lottery probabilities are relatively high, individuals have been shown to be mostly risk-averse in the gain frame and risk-seeking in the loss frame. Note, however, that Prospect Theory is silent about the effects of time pressure on such valence-dependent pattern. Our interpretation is, thus, that safe and risky options are *natural default* options in, respectively, gains and loss contexts, and that time pressure increases these natural default options biases. This interpretation is in line with a recent study proposing that individuals frame choices relatively to prior preferences for certain categories of options [56].

The novel contribution of the current study, which follows from the careful control over multiple experimental factors, is the first dissociation of several potential dimensions of endogenous default options (natural, dominant and learned default options, **Fig 1**, see also **S1 File**, **Additional Behavioral Predictions**), all of which could be–at first sight–linked to an automatic and fast process, a very general umbrella term under dual-process theory assumptions. These dimensions, although potentially underpinned by different neuro-cognitive processes, could be confounded under simpler experimental designs, and lead to mis-interpretations and apparently contradictory results. In our experiment, we explicitly manipulated choice proportion by making one class of options–the risky or the safe options–more desirable, hence more likely to be chosen. We derived the properties of our lotteries in accordance with a pre-specified average probability (70%) of choosing the desired class option and with expected utility model-parameters obtained in a preliminary experiment (see **S1 File**, **Supplementary Experiments**). The rationale was that time pressure could increase the proportion of the most frequently chosen option or simply choice repetition (*dominant default*). Under this hypothesis, inconsistencies between results of different studies could be explained by differences in the lottery set composition, and/or in the average risk preferences observed in the sample of participants. Yet, ruling out this possibility, we found that time pressure effects on risky choices in gain and loss domains were independent of the quality of the most frequently chosen option (risky or safe). In this experiment, we also paid special attention to the time on task factor. The time on task factor is often neglected and/or not analyzed, although present in all empirical studies. It can correlate with the learning (implicit or explicit) of task contingencies, efficient strategies and heuristics, and to increasing boredom, mind wandering or fatigue [57,58]. In contrast to time pressure effects, our results show that choice proportion interacts with the time on task: while participants progressed through the experiment, they increasingly chose the most frequently chosen option. In other words, with increasing time spent on task, participants learn implicitly–i.e., without feedback–which option they choose more frequently, and

this biases their choices (*learned default*). Although probability distortions are not considered in our model to generate lotteries (see **Stimuli**), it would be of interest for delineating and refining the actual properties of defaults options if future studies investigate the effects of such probability distortions with a slightly different stimuli set [59]. The notion of endogenous default options has been explicitly linked with dual-process theories, with the default option being preferentially endorsed and chosen by a fast and automatic process, and the alternative options by a slow, deliberative one [17]. Although theoretically appealing, dual-process theories have also been criticized for being overly simplistic and lacking unambiguous experimental support due to somewhat vague theoretical specifications [60–62]. Here, we report the results of a statistical test of a quite conservative specification of dual-process hypothesis: the fixed-point property test of RT distributions [35,48]. This specification states that, assuming that two processes can execute the decision in a strictly independent manner with distinct RT distributions, the observed distribution of RTs is a mixture of those distributions, and all mixture proportions should share a common density point. Consistent with prior work in perceptual decision-making [63,64], we found no evidence for a fixed-point under the different levels of time pressure. Note, however, that the fixed-point test is a conservative test, with conservative assumptions: if, in addition to the change in the mixture proportion, time pressure affects some other aspect of the data then the fixed-point property may not apply, even though the actual true generative process is a mixture of processes [35,36]. An example of this is when the level of response caution decreases resulting in shorter RTs [65–68], or if the natural default is privileged when the expected-value computation time is too long [69,70]. Likewise, it is unclear how some specifications of dual-process theories–like the default-interventionist model which assumes a take-over of the deliberative process over the automatic process under certain conditions [17]–would satisfy the fixed-point assumption about independent processes. Still, the absence of evidence for a fixed-point property across blocks of time pressure suggests that the natural default effects may rather be explained as a preference-induced bias in the computations leading to the decisions [56,65,71], than as a consequence of a fast, intuitive and automatic process. Despite the stringent assumptions underlying the fixed-point test, we found evidence for a fixed-point property in RT distributions across blocks of time on task. This suggests that the emergence of the learned default is due to the emergence of an independent decision-process, like an explicit strategy or a heuristic-like process [72], for which individuals apply 'simple rules'–here, in the form of simple repetition of the most frequent choice type–that they learn implicitly through experience, and through the course of the task. Future studies investigating the effects of these default options using model-based approaches (e.g., sequential sampling models), would be a good complement to the fixed-point property analysis. Here, we focused on the fixed-point property, because it specifically tests hypotheses about single versus dual processes–an outstanding question with respect to the link between default options and dual-process models of cognition.

Overall, our results demonstrate that systematic investigations of the multiple dimensions of endogenous default options provide valuable insights into the origin and mechanisms of decision bias observed in humans. We argue that such systematic investigations are extremely important to overcome the criticisms that dual-process theories have been targeted of. These dimensions, although potentially underpinned by different neuro-cognitive processes, could be confounded under simpler experimental designs, and lead to mis-interpretations and apparently contradictory results. Importantly, these dimensions can provide explicit testable predictions regarding the neural coding of decision variables [56,73]. Whether default options similar to the ones we describe in this study are present in non-human primates or other animals is an outstanding question. For example, it has been shown that non-human primates attitude towards risky choices can change with trial sequence repetition [59]. Yet, we are aware

that experimental attempts to replicate or extend the present results in non-human animals might face important challenges. If, on one hand, in animal studies all decisions have to be followed by outcomes (i.e., rewards), on the other hand, the presence of feedback after choices has been repeatedly shown to elicit different decisions patterns in humans–a phenomenon called description-experience gap [74,75].

## Supporting information

**S1 File. Supplementary information.** First section describes Experiment S1 and Experiment S2, preliminary experiments of Experiment 1 and 2, respectively. Second section provides supplementary information on Experiment 1 and 2 (i.e., on additional behavioral predictions, stimuli generation, GLMMs).
(DOCX)

## Acknowledgments

We thank Kim Archambeau and Ivar Kolvoort for their insightful comments on the manuscript.

## Author Contributions

**Conceptualization:** Joaquina Couto, Leendert van Maanen, Maël Lebreton.

**Formal analysis:** Joaquina Couto, Leendert van Maanen, Maël Lebreton.

**Investigation:** Joaquina Couto.

**Methodology:** Joaquina Couto, Leendert van Maanen, Maël Lebreton.

**Project administration:** Leendert van Maanen, Maël Lebreton.

**Writing – original draft:** Joaquina Couto, Leendert van Maanen, Maël Lebreton.

**Writing – review & editing:** Joaquina Couto, Leendert van Maanen, Maël Lebreton.

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
