## [Decision Letter · Decision Letter 0]

20 May 2020

PONE-D-20-10214

Investigating the origin and consequences of endogenous default options in repeated economic choices.

PLOS ONE

Dear Mrs Couto,

Thank you for submitting your manuscript to PLOS ONE. After careful consideration, we feel that it has merit but does not fully meet PLOS ONE’s publication criteria as it currently stands. Therefore, we invite you to submit a revised version of the manuscript that addresses the points raised during the review process.

We would appreciate receiving your revised manuscript by Jul 04 2020 11:59PM. To enhance the reproducibility of your results, we recommend that if applicable you deposit your laboratory protocols in protocols.io, where a protocol can be assigned its own identifier (DOI) such that it can be cited independently in the future. For instructions see: http://journals.plos.org/plosone/s/submission-guidelines#loc-laboratory-protocols

We look forward to receiving your revised manuscript.

Kind regards,

Alireza Soltani

Academic Editor

PLOS ONE

Journal Requirements:

1. Please ensure that your manuscript meets PLOS ONE's style requirements, including those for file naming. The PLOS ONE style templates can be found at https://journals.plos.org/plosone/s/file?id=wjVg/PLOSOne_formatting_sample_main_body.pdf andhttps://journals.plos.org/plosone/s/file?id=ba62/PLOSOne_formatting_sample_title_authors_affiliations.pdf

2. Please include a caption for figure 5.

Reviewers' comments:

Reviewer's Responses to Questions

**Comments to the Author**

1. Is the manuscript technically sound, and do the data support the conclusions?

Reviewer #1: Yes

Reviewer #2: Yes

2. Has the statistical analysis been performed appropriately and rigorously? 

Reviewer #1: Yes

Reviewer #2: Yes

3. Have the authors made all data underlying the findings in their manuscript fully available?

Reviewer #1: No

Reviewer #2: Yes

4. Is the manuscript presented in an intelligible fashion and written in standard English?

Reviewer #1: Yes

Reviewer #2: Yes

5. Review Comments to the Author

Reviewer #1: In the manuscript “Investigating the origin and consequences of endogenous default options in repeated economic choices”, Couto and colleagues aim to investigate the existence and properties of endogenous default options. Authors adopt a standard economic binary choice task with varying monetary amounts and probabilities of gains or losses. By probing the effects of three experimental features including time pressure, time spent on task and relative choice proportion on, authors dissociated between two features of endogenous default options: natural default (a natural tendency to prefer certain types of options) and learned default (the tendency to implicitly learn a default option from past choices). Moreover, authors show that the natural default option bias results from a distortion of the decision-making process under time pressure, while the learned default option bias results from the emergence of an independent process endorsing the most frequently chosen option with increasing time on task.

The manuscript is well written and describes an interesting question about endogenous default options in economic choices. I have a few suggestions and a major comment on the effect of ‘learned default option’ that should be addressed to strengthen the conclusions of the paper. Please find my comments below:

Major concerns:

- Page 9, line 213; Is there any reason why authors do not consider a probability weighting function in their fit?

- Related to PWF, a previous study on primates’ choice behavior [1], has shown a change in probability distortion function as a result of repeated choice sequence. How is the ‘learned default option’ different than a change in PWF that can happen as a result of over-exposure to a certain type of options? How does a change like that effect authors claims? Please comment on this.

[1]. Ferrari-Toniolo, S., Bujold, P. M., & Schultz, W. (2019). Probability distortion depends on choice sequence in rhesus monkeys. Journal of Neuroscience, 39(15), 2915-2929.

Minor concerns:

- To increase readability, I suggest the authors to add their explanation of fixed-point property to the introduction section. This way, the introduction section includes all the required information to understand the authors’ claims and results in the abstract.

- I would recommend authors to use ‘experience’ or some other terminology to refer to the effect of experience with task. The phrase ‘time spent on task’ was a bit confusing to me. Without the explanation in the main text, I thought authors are referring to reaction time in choice trials where there in no time pressure.

- I suggest that authors show ‘No Default’ case first instead of last in Fig. 1.

- Please avoid using top/bottom to refer to figures. Authors should consider adding labels to Fig. 1 bottom row and Fig 4B.

- Please add a sentence explaining what asterisks represent in Fig. 2 and Fig. 5.’s captions.

- Page 17, line 418; It would be helpful to provide a figure or a statistical comparison of fixed-point analysis for Experiments 1 and 2, confirming the assumption that there is no difference between the two experiments.

Reviewer #2: This paper studied the existence and properties of endogenous default options in risky decision making. To do that, the authors manipulated and investigated the effects of three experimental features, time pressure, time spent on task and relative choice proportion towards a specific option. They found and dissociated two features of endogenous default options which could bias individuals’ choices, the natural default and the learned default. Moreover, the natural default biases the choice process towards an option category, while the learned default effects may be attribute to another process.

This study is interesting, but has the following problems:

(1) I am a little bit confused about the term “default”. Is it really the default option or a prior bias? How do you know it is the default option?

(2) The authors related the default to the automatic or fast process in dual-process theories, is the automatic process the default process? This is also related to the question above. Is it a default bias, or an automatic bias, or a prior bias?

(3) In the abstract, the authors mentioned that “the learned default effects may be attributable to a second independent choice process.”What is the “second independent choice process”? Could you please make it more explicitly?

(4) The “dominant default” is induced by the mere repetition of the most frequent choice type, and the “learned default” is induced over time by the repetition of the most frequent choice type. In this sense, are these two default endogenous or exogenous?

(5) Figure 1 needs more explanations. It is not clear how the authors get the figure.

(6) “Top panels” and “Down panels” in Figure 3 should be “Left panels” and “Right panels”?

(7) “The results show that participants predominantly chose the option designed to be the most frequently chosen; participants increasingly chose the risky option with increasing time pressure, regardless of the most frequently chosen option; and participants increasingly chose the most frequently chosen option while they progressed through the experiment… “All these results are at the aggregate/group level, is there any individual differences? For instance, most of them go to the same direction, but others go to the opposite direction.

(8) More explanations are needed for the fixed-point property. Why the fixed-point property exists when the observed data are the result of a mixture of two base distributions, and why not for a single process?

(9) In Figure 4A, the authors mentioned the red dot indicates the fixed-point, but there is no red dot in Figure 4A.

(10) Studies have used sequential sampling models to investigate the process underlying binary decisions and the biases in the decision process. Why the authors do not use sequential sampling models?

6. PLOS authors have the option to publish the peer review history of their article (what does this mean?). If published, this will include your full peer review and any attached files.

Reviewer #1: No

Reviewer #2: No

---

## [Author Response · Author response to Decision Letter 0]

17 Jun 2020

Dear Dr. Alireza Soltani,

Thank you for considering our manuscript for resubmission in PLOS ONE. We also thank the reviewers for all their constructive feedback and insightful comments. We thereby resubmit a revised version, along with all the required items: 

• The rebuttal letter (see attached ‘Response to Reviewers’ ), where we reply to every reviewers' comments in detail.

• The marked-up copy of the revised manuscript (see attached ‘Revised Manuscript with Track Changes’) highlighting the changes we made based on reviewers’ comments.

• The unmarked version of the revised manuscript (see attached ‘Manuscript’) without tracked changes.

In addition, and in response to reviewers’ comments and/or editors’ requests please note that:

• The revised manuscript has been updated according to PLOS ONE's style requirements, including file naming (see attached ‘Fig1’, ‘Fig2’, ‘Fig3’, ‘Fig4’, ’Fig5’,’Fig6’, ‘S1_File’).

• Captions for Fig 4 and 5 are added and caption for Fig 4 is updated accordingly (now as Fig 6).

• Captions for Supporting Information files are added in the end of the revised manuscript and in-text citations are updated accordingly.

• All data are fully available without restriction. Please check the data availability section of the submission. Specifically, all data are fully available in an OSF repository, DOI 10.17605/OSF.IO/TSJBU.

Sincerely,

Joaquina Couto

---

## [Decision Letter · Decision Letter 1]

17 Jul 2020

Investigating the origin and consequences of endogenous default options in repeated economic choices.

PONE-D-20-10214R1

Dear Dr. Couto,

We’re pleased to inform you that your manuscript has been judged scientifically suitable for publication and will be formally accepted for publication once it meets all outstanding technical requirements.

Kind regards,

Alireza Soltani

Academic Editor

PLOS ONE

Reviewers' comments:

Reviewer's Responses to Questions

**Comments to the Author**

1. If the authors have adequately addressed your comments raised in a previous round of review and you feel that this manuscript is now acceptable for publication, you may indicate that here to bypass the “Comments to the Author” section, enter your conflict of interest statement in the “Confidential to Editor” section, and submit your "Accept" recommendation.

Reviewer #1: All comments have been addressed

Reviewer #2: All comments have been addressed

2. Is the manuscript technically sound, and do the data support the conclusions?

Reviewer #1: Yes

Reviewer #2: Yes

3. Has the statistical analysis been performed appropriately and rigorously? 

Reviewer #1: Yes

Reviewer #2: Yes

4. Have the authors made all data underlying the findings in their manuscript fully available?

Reviewer #1: Yes

Reviewer #2: Yes

5. Is the manuscript presented in an intelligible fashion and written in standard English?

Reviewer #1: Yes

Reviewer #2: Yes

6. Review Comments to the Author

Reviewer #1: I would like to thank the authors for addressing all my concerns. I have no further questions/comments.

Reviewer #2: (No Response)

7. PLOS authors have the option to publish the peer review history of their article (what does this mean?). If published, this will include your full peer review and any attached files.

Reviewer #1: No

Reviewer #2: No

---

## [Editor Report · Acceptance letter]

3 Aug 2020

PONE-D-20-10214R1 

Investigating the origin and consequences of endogenous default options in repeated economic choices. 

Dear Dr. Couto:

I'm pleased to inform you that your manuscript has been deemed suitable for publication in PLOS ONE. Congratulations! Your manuscript is now with our production department. 

Kind regards, 

on behalf of

Dr. Alireza Soltani 

Academic Editor

PLOS ONE